# Energy-efficient UAV communication: A NOMA scheme with resource allocation and trajectory optimization

**Huilong Jin[1,2], Yucong Zhou[1], Xiaozi Jin[1], Shuang Zhang** [1,2]*

**1** Hebei Normal University, Shijiazhuang, China, **2** Hebei Provincial Key Laboratory of Information Fusion and Intelligent Control, Shijiazhuang, China

* zshuang@hebtu.edu.cn

## Abstract

This work investigates a downlink nonorthogonal multiple access (NOMA) scheme with unmanned aerial vehicle (UAV) aided wireless communication, where a single UAV was regarded as an air base station (ABS) to communicate with multiple ground users. Considering the constraints of velocity and maneuverability, a UAV energy efficiency (EE) model was proposed via collaborative design resource allocation and trajectory optimization. Based on this, an EE maximization problem was formulated to jointly optimize the transmit power of ground users and the trajectory of the UAV. To obtain the optimal solutions, this nonconvex problem was transformed into an equivalent convex optimization problem on the basis of three user clustering algorithms. After several alternating iterations, our proposed algorithms converged quickly. The simulation results show an enhancement in EE with NOMA because our proposed algorithm is nearly 99.6% superior to other OMA schemes.

## 1 Introduction

### 1.1 Background and motivation

In recent decades, vast natural disasters, such as earthquakes, wildfires, and tsunamis, have caused serious damage to communication equipment and hampered the normal operation of communication networks. Postdisaster situational awareness is urgently needed to be obtained by maintaining real-time communications, which can vastly improve the efficiency of rescue missions [1]. Unmanned aerial vehicles (UAVs) are capable of serving as prospective communication platforms due to their flexibility, mobility, and cost effectiveness [2]. Therefore, establishing a multitudinous emergency communication network by deploying UAVs in a timely manner after disasters. For instance, when a communication network is partially or completely destroyed, a UAV can connect with responsive personnel first and act as a coverage heightening relaying node [3]. UAVs have been broadly used in various natural disaster management applications [2]; UAVs contribute to addressing complex ground conditions and insufficient electrical supplies after disasters.

This approach can achieve benefits in terms of spectrum efficiency for UAV-aided systems. We focus on nonorthogonal multiple access (NOMA), which has been deemed a prospective

**Funding:** This research was funded by the following awards: 1.Industry-University-Research

Innovation Foundation of Chinese University (2021LDA06003,URL:http://www.cutech.edu.cn/cn/index.htm), whose project leader is Huilong Jin. The sponsors play a role in logic structure and writing skill of this manuscript. 2.Science and Technology Research Foundation of Hebei Normal University (L2021B33,URL: https://www.hebtu.edu.cn/) whose project leader is Shuang Zhang. The sponsors play a role in direction of application.

**Competing interests:** The authors have declared that no competing interests exist.

multiple access (MA) technology to yield a remarkable spectral efficiency gain in communication networks [4]. In contrast to previous generations of orthogonal multiple access (OMA), which depend on the spectrum/time/code domain [5], NOMA achieves high spectral efficiency by incorporating superposition coding at the transmitter with successive interference cancellation (SIC) at the receivers. This provides an effective pathway for UAVs to meet different needs for power from massive numbers of ground users. In addition, NOMA primarily leverages diverse channel gain differences to superpose the message signals of multiple users with varying transmission powers at the transmitter, facilitating simultaneous frequency transmission [6]. At the receiver, serial interference cancellation techniques are employed to demodulate user information [7]. This work specifically focuses on line-of-sight (LoS) communication between UAVs and ground users, where the transmission quality is considerably superior to that of non-line-of-sight (nLoS) transmission between ground users and base stations. Implementing NOMA in both scenarios further emphasizes the spectral efficiency advantage of NOMA.

However, the limited battery life of UAVs poses a considerable challenge for incorporating joint NOMA technology into communication applications. Therefore, energy efficiency (EE) is one of the key problems in UAV communication with NOMA [8]. Based on our previous research, four methods are summarized: EE in trajectory planning and deployment, resource allocation and management, energy harvesting and transfer, and communication protocol design [9]. The main objective is to reduce energy consumption and cost by focusing on the former two methods. Owing to the size and weight constraints of aircraft, the performance and durability of UAVs are radically limited by onboard energy storage [10]. Hence, communication tasks should be completed as much as possible before the UAV exhausts energy and within the user's equipment usage time. This work aims to address the problem of EE maximization and further improve system performance. Therefore, three user clustering algorithms and a maximum EE algorithm are considered to improve the EE by optimizing the trajectory and resource allocation in downlink NOMA. Moreover, the energy consumption related to EE can be divided into two categories: (i) conventional energy consumption related to communication and (ii) additional propulsion energy consumption [9, 11]. The details are explained in Section II. The propulsion energy consumption for maintaining a UAV aloft and supporting its mobility (if necessary) is usually much greater than the communication power consumption; thus, the flying status of a UAV, including velocity, acceleration and orientation, is needed to reduce energy consumption [11].

## 1.2 Related work and contribution

Many studies have analyzed UAV-aided wireless communication, and the problem of UAV EE has attracted the attention of scholars in the past few years. Several articles have investigated the factors influencing EE, including trajectory planning, resource allocation, and height, in UAV communication. Recent surveys are reviewed in detail and supplemented with other relevant work.

First, considering the high mobility of UAVs, we focus on trajectory design and optimization, which provides a fresh degree of freedom for optimizing the performance of wireless communication systems. Previous studies have investigated the problem of trajectory optimization by adjusting various setups [12–17]. To maximize the number of covered users, the placement problem was translated into a plural circle placement problem to explore the UAV trajectory [12]. To obtain the maximal throughput gain, user scheduling and spectrum resource allocation were jointly optimized in [13], and the design of trajectory, transmission power and communication scheduling regarding UAVs were similarly investigated in [14]. To

minimize energy consumption, the authors of [15–17] contributed to three-dimensional (3D) trajectory optimization. [15] proposed a revised genetic algorithm (GA) to optimize the trajectory. A new energy consumption model was derived to solve the trajectory optimization problem in UAV-aided wireless communication in [16]. Unlike previous jobs that relied on only kinematic equations, a control-based trajectory according to both kinematic equations and dynamic equations of UAVs was designed in [16]. The UAV autonomously determines its trajectory on the basis of the reinforcement learning framework, which reduces the energy consumption of the UAV [17]. Although the problem of reducing energy consumption was considered, the works above did not extend to EE performance.

The relevant methods for resource allocation are described in this section. A suboptimal resource allocation scheme, including user scheduling and power allocation, was designed with NOMA in [18]. Similarly, a problem with resource allocation was formulated by jointly optimizing subchannel selection and uplink transmission power control in an internet of remote things network in [19]. As a supplement to previous work, a trajectory based on the successive convex approximation (SCA) scheme was further proposed in [20], where the joint optimization of resource allocation and trajectory was considered. Similarly, to maximize the data uploading throughput, the joint design of trajectory, transmission power and communication scheduling for UAVs was studied in [14]. The placement, admission control and power allocation were jointly designed to maximize the number of users with satisfactory QoS experience in [21]. However, the optimization of the EE for UAVs was not considered in these works. Several studies improved upon this approach [22–25], where optimal solutions exploiting the SCA with rapid convergence were obtained. A matching and swapping algorithm was used to solve the EE problem for UAVs with NOMA in each subperiod [22]. Otherwise, in light of a three-layer iterative algorithm, the user scheduling problem can be optimally solved with low computational complexity and EE for any given UAV trajectory and transmit power [23]. In contrast to [23], the joint design of EE in [24] was formulated as a nonconvex optimization problem considering its minimum data rate requirement, the minimum safety distance between UAVs, and the imperfect location information of potential eavesdroppers. In addition, a pattern of UAV-assisted mobile edge computing was studied in depth to improve computational offloading and EE by optimizing the trajectory, transmit power and computational load allocation [25]. In conclusion, there exists interdependence and mutual influence between resource allocation and trajectory planning. By considering comprehensive metrics that incorporate both resource allocation and trajectory planning, UAVs can efficiently accomplish communication tasks and achieve optimal EE under limited resource conditions. This paper aims to improve the performance of UAV communication systems based on joint optimization.

Moreover, UAVs are energy-constrained, and ground users face the challenge of interference from others who share the same frequency spectrum, which influences the EE of the system. Recent approaches for EE focus on dividing users into multiple clusters in NOMA. Each cluster contains a certain number of users and scheduling resources, which ensures that there is no interference among users. The sum rate is maximized by a novel scheme that jointly optimizes the UAV trajectory and the NOMA precoding in [7], which eliminates the interference from the BS to the UAV-served user. Nevertheless, the growing number of ground users has increased the complexity and decoding time in practical scenarios for SIC. The performance gain of F-NOMA can be further expanded by selecting users who have distinctive channel conditions [26]. On this basis, a low-complexity suboptimal user grouping scheme was proposed in [27], where users are divided into an NOMA cluster based on their different gains. Then, [28] proposed an improved user pairing strategy to increase the capacity gain for almost all users. The simulation results showed that the schemes achieved greater

capacity gains, especially when imperfect SIC was considered [28]. In addition, [29] designed a power allocation strategy by applying the user clustering methods mentioned above. However, the performance of EE was not considered in these works. Therefore, this paper proposes a joint optimization scheme with UAV trajectory and transmission power to maximum EE. Different from the works in [23–25], we additionally consider the situation of ground users and innovatively design a user clustering algorithm based on the Dinkelbach method. Table 1 shows a summary of related work. The main contributions of the paper are summarized as follows:

- This paper considers a UAV-assisted communication system where a single UAV is viewed as an ABS connecting multiple ground users with NOMA. A theoretical model of propulsion energy consumption connected with flying velocity is formulated, based on which, a corresponding nonconvex problem of EE is proposed.

- To fully exploit the channel gains in NOMA, three types of user clustering algorithms are designed with different grouping methods: greedy clustering, suboptimal clustering and hybrid clustering algorithms. The simulation results indicate that different user clustering algorithms have different influences on the EE of UAV systems. The hybrid clustering algorithm is more effective than the other two algorithms.

- To obtain the optimal solution, we transform the original problem into a convex optimization problem. A maximum EE algorithm focused on the resource allocation scheme is proposed to solve the optimization problem by exploiting the Dinkelbach method. The simulation results show that the proposed algorithm is superior to other benchmarks.

**Table 1. Related works.**

| Paper | Approach | CC Partitioning | Trajectory | Resource Allocation | Interference | Main idea |
|---|---|---|---|---|---|---|
| [12] | ES,MWA | – | 3D | × | × | Coverage |
| [13] | SCO,ADMM | – | 3D | ✓ | ✓ | Throughput |
| [14] | SCA,RHO,KKT | – | 2D | ✓ | × | |
| [15] | EFDC,TSP,GA | – | 3D | × | × | Energy consumption |
| [16] | state-space,dynamic | – | 3D | ✓ | × | |
| [17] | RL,MDP | – | 3D | ✓ | × | |
| [21] | SCA,KKT,NOMA,SIC | – | 2D | ✓ | ✓ | User number |
| [18] | SCA,ES,DC programming | – | × | ✓ | ✓ | Improved EE by power and outage probability |
| [19] | SCA,RA-CPD algorithm | – | × | ✓ | × | Improved EE by power and sub-channel allocation |
| [20] | SCA | – | 2D | ✓ | ✓ | Improved EE by power and trajectory |
| [22] | SCA,MS algorithm, | – | 2D | ✓ | × | |
| [23] | SCA,Dinkelbach | – | 2D | ✓ | ✓ | |
| [24] | SCA,Dinkelbach | – | 2D | ✓ | × | |
| [25] | SCA,Dinkelbach | – | 2D | ✓ | ✓ | |
| This work | User clustering algorithm, Dinkelbach | K-Clusters | 3D | ✓ | ✓ | |
| [26] | Gale-Shapley matching, game theory | K-Groups | 3D | × | ✓ | Sum rate |
| [27] | Sub-optimal user grouping algorithm | K-Groups | × | ✓ | ✓ | Throughput |
| [28] | User pairing algorithm | K-Pairs | × | × | ✓ | Capacity |
| [29] | Interval grouping algorithm | K-Groups | × | ✓ | ✓ | |

The rest of the paper is organized as follows. Section 2 introduces the system model in detail; We formulated the problem named maximization of EE in section 3; In section 4, the user clustering algorithm is given to optimize the EE. Simulation results are discussed in Section 5. Finally, section 6 summarizes the work of this paper.

## 2 System model

### 2.1 Deployment model

The scenario of interest is examined, and a 3D system model is considered. A UAV-supported NOMA system with a single UAV providing data services is used as an air base station for multiple users in this paper. In the 3D Cartesian coordinate system, the flight altitude of the UAV during the flight cycle $T$ is assumed to be fixed to $H$, and the horizontal coordinate is set as $q(t) = [x(t), y(t)]^T$, $0 \leq t \leq T$. Assume that the UAV flies over the area of interest and offers service for $U$ users at the same time, $U \in N^+$.

As shown in Fig 1, $U$ users are divided into $K$ clusters, and each cluster consists of $N$ users, where $K, N \in N^+$, $U = KN$. The set of clusters is $\mathcal{K} = \{1, 2, \cdots, N\}$. Thus, the $n$-th user in the $k$-th cluster is denoted as $U_{k,n}$. Consider a 3D coordinate system, where the altitude of the user

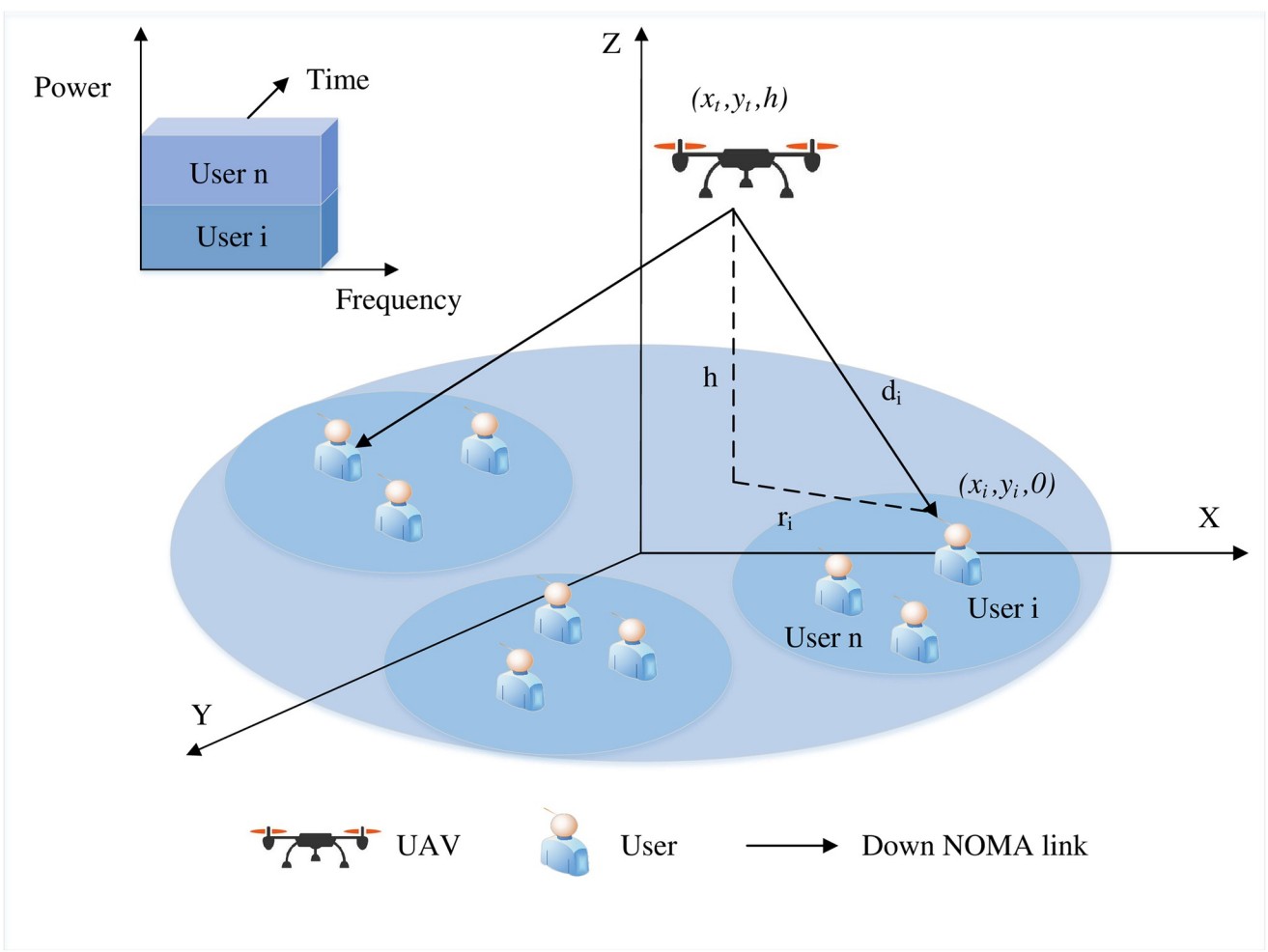

**Fig 1. System model.**

**Table 2. Table of notations.**

| Notation | Definition | Notation | Definition |
|---|---|---|---|
| $U$ | number of total users | $K$ | number of user clusters |
| $N$ | number of user in each cluster | $H$ | flight altitude of UAV |
| $T$ | flight cycle of UAV | q | horizontal coordinate of UAV |
| v | speed of UAV | p | transmission power vector of users |
| $B$ | channel bandwidth | $U_{k,n}$ | $n$th user in the $k$th cluster |
| $W_{k,n}$ | horizontal coordinate of $U_{k,n}$ | $h_{k,n}$ | channel power gain between $U_{k,n}$ and UAV |
| $d_{k,n}$ | 3D distance between UAV and $U_{k,n}$ | $y_{k,n}$ | received signal at the user $U_{k,n}$ |
| $n_{k,n}$ | additive white Gaussian noise | $SINR_{k,n}$ | received SINR at the $n$th user in $k$th cluster |
| $R_{k,n}$ | transmission rate for $U_{k,n}$ | $R_{total}$ | total transmission rate for all users |
| $R_{target}$ | goal of transmission rate | $\bar{E}$ | total energy consumption |
| $EE, \eta$ | energy efficiency of UAV system | $P_{\max}$ | maximum transmission power of UAV |
| $G_0, G_1$ | index of the directional antenna gains | $I, o$ | iteration index |

is regarded as $0m$ and the horizontal coordinate of $U_{k,n}$ is denoted as $W_{k,n} = [x_{k,n}, y_{k,n}]^T, \forall k, n$. Suppose that the UAV and all users are mounted with one single antenna. The notations and their definitions are given in Table 2.

## 2.2 NOMA rate model

The objective is to enable data transmission between a UAV in the air and users on the ground. Consequently, the widely adopted air-to-ground channel model is utilized, where the communication links are LoS to capture the distortion of the signal due to obstructions. Furthermore, the Doppler effect due to UAV mobility is assumed to be ideally compensated for. Thus, the channel between user $U_{k,n}$ and the UAV follows the free-space path loss model, which can be expressed as

$$h_{k,n}(t) = \frac{G_0 G_1 \beta_0}{d_{k,n}(t)^2}, \forall k, n, \tag{1}$$

where $G_0$ and $G_1$ represent the directional antenna gains of the UAV and users, respectively. $\beta_0$ represents the channel power gain at the reference distance $d_0 = 1m$. $d_{k,n}$ is the distance between the UAV and user $U_{k,n}$, i.e.,

$$d_{k,n}(t) = \sqrt{H^2 + \|q(t) - W_{k,n}\|^2}, \forall l, k, n, \tag{2}$$

where $\sqrt{\|q(t) - W_{k,n}\|^2}$ represents the horizontal distance between the UAV and the users.

We consider the downlink communication based on NOMA; then, the received signal at user $U_{k,n}$ can be expressed as [1]

$$y_{k,n}(t) = h_{k,n}(t)\bar{x}_{k,n} + h_{k,n}(t)\left(\sum_{j=1,j\neq n}^{N} \bar{x}_{k,j} + \sum_{i=1,i\neq k}^{K} \sum_{j=1}^{N} \bar{x}_{i,j}\right) + n_{k,n}. \tag{3}$$

$\bar{x}_{k,n}$ is the transmitted message for user $U_{k,n}$. $n_{k,n}$ represents the additive white Gaussian noise (AWGN) following $\mathcal{CN}(0, \sigma^2)$. $\left(\sum_{j=1,j\neq n}^{N} \bar{x}_{k,j} + \sum_{i=1,i\neq k}^{K} \sum_{j=1}^{N} \bar{x}_{i,j}\right)$ represents the sum of the transmitted messages for all users except for the $k$-th cluster's $n$-th user. Without loss of generality, the channel gains in the $k$-th cluster follow $0 < \|h_{k,1}(t)\|^2 \leq \cdots \leq \|h_{k,n}(t)\|^2 \leq \cdots \leq \|h_{k,n}(t)\|^2$. $P_{\max}$ is defined as the maximum transmission power of the UAV, and p = $(p_{1,1},$

$p_{1,2}, \cdots, p_{k,n}, \cdots, p_{k,n}$) is the transmission power vector of the users. According to NOMA, SIC is utilized by users according to their channel conditions [30]. For example, user $U_{k,n}$ must decode information from the $1st$ to the $(n-1)$th user in the $k$-th cluster before decoding its own information. Then, the received signal-to-interference-plus-noise ratio (SINR) at the $n$-th user complying with $(1 \leq n \leq N-1)$ in the $k$-th cluster can be presented as

$$\text{SINR}_{k,n}(q(t)) = \frac{p_{k,n}}{\sum_{i=n+1}^{N} p_{k,i} + \alpha(H^2 + \|q(t) - W_{k,n}\|^2)} \tag{4}$$

with a function of the $i$-th trajectory $q(t)$, where $\alpha = \frac{\sigma^2}{G_0 G_1 \beta}$. Specifically, the SINR is the ratio of the strength of the received useful signal to the strength of the received interfering signal (noise and interference). For the $K$-th user, the received SINR can be denoted by

$$\text{SINR}_{K,N}(q(t)) = \frac{p_{K,N}}{\alpha(H^2 + \|q(t) - W_{K,N}\|^2)}. \tag{5}$$

According to the definition, the transmission rate for user $U_{k,n}$ in bits/second can be presented as

$$R_{k,n}(q(t)) = B \log_2 (1 + SINR_{k,n}(q(t))), \tag{6}$$

where $B$ is the channel bandwidth. Thus, the total transmission rate for all users can be expressed as

$$R_{total}(q(t)) = B \sum_{k=1}^{K} \sum_{n=1}^{N} \log_2 (1 + \text{SINR}_{k,n}(q(t))). \tag{7}$$

$\bar{R}_{total}$ represents the total amount of information bits, which is a function of the UAV trajectory $q(t)$ over the duration $T$ in Eq (8).

$$\bar{R}_{total}(q(t)) = B \sum_{k=1}^{K} \sum_{n=1}^{N} \int_0^T \log_2 \left(1 + \frac{p_{k,n}}{\sum_{i=n+1}^{N} p_{k,i} + \alpha(H^2 + \|q(t) - W_{k,n}\|^2)}\right) dt. \tag{8}$$

## 2.3 Energy efficiency model

The energy consumption of the UAV communication includes two components. The first is communication-related energy consumption, which is caused mainly by radiation, signal processing and other circuits. The other component is propulsion energy consumption, which is required for ensuring that the UAV remains hovering and moves aloft. Notably, communication-related energy consumption is usually ignored in practical scenarios because it is much lower than the propulsion energy consumption of UAVs.

Assume that the UAV flies in a horizontal direction with a stationary altitude $H$, which corresponds to the minimum altitude required to avoid obstacles in the region. For a fixed-wing UAV under normal operation, i.e., with no abrupt deceleration that requires the engine to abnormally produce a reverse thrust against the forward motion of the aircraft, the total propulsion energy required is a function of the instantaneous coordinates with uniform motion $V = \|v\|^2$ [29], which is expressed as

$$\bar{E}(q(t)) = \int_0^T \left[c_1 \|v(t)\|^3 + \frac{c_2}{\|v(t)\|}\right] dt, \tag{9}$$

where $c_1$ and $c_2$ are two parameters related to the UAV's weight, wing area, air density, etc. In

addition, there are sets

$$\mathrm{v}(t) \triangleq \dot{q}(t), \tag{10}$$

where $\dot{q}(t)$ denotes the derivative with respect to time $t$. In principle, Eq (9) represents the work done by the engine to overcome the air resistance during UAV flight. For Eq (9), it indicates that the energy consumption of a UAV depends on the velocity $\mathrm{v}(t)$ at a fixed altitude.

Thus, according to Eqs (8) and (9), the EE of the UAV communication can be expressed as

$$EE(\mathrm{q}(t)) = \frac{\bar{R}_{total}(\mathrm{q}(t))}{\bar{E}(\mathrm{q}(t))}. \tag{11}$$

## 3 Problem formulation

With the constraints of the achievable sum rate and mobility of the UAV in the downlink, the aim of this paper is to maximize the EE in the downlink by synergistically designing the UAV trajectory and transmission power for users. In addition, the influence of user clustering algorithms on the EE is considered. Therefore, the problem is expressed as

$$
\begin{aligned}
(\mathrm{P}\,1)\colon &\max_{\mathrm{q}(t)} EE(\mathrm{q}(t)) \\
\text{s.t. } &\mathrm{C}\,1 : \|R_{total}(\mathrm{q}(t))\| \geq R_{target} \\
&\mathrm{C}\,2 : p_{k,n} < P_{\max} \\
&\mathrm{C}\,3 : \sum_{i=1}^{N} p_{k,i} \leq P_{\max}\ ,
\end{aligned}
\tag{12}
$$

where $R_{target}$ denotes the goal of the transmission rate. Restriction (12-C1) ensures that the transmission rate of all users is not lower than the target sum rate. (12-C2) ensures that the transmission power of each user is no more than the transmission power of the UAV; (12-C3) guarantees that the transmission power of each user cluster is no more than the transmission power of the UAV. In practice, mathematical problem (12) involves a nonconvex optimization problem, which is exceedingly difficult to solve directly. Consequently, we reconstruct (P1) to simplify the problem.

By discretizing the time horizon $T$ into $L + 2$ slots with step size $\tau_t$, i.e., $l = 0, 1, \cdots, L + 1, l = n\tau_t$, the UAV's trajectory $\mathrm{q}(t)$ can be well characterized by the discrete-time UAV $\mathrm{q}[n] \triangleq \mathrm{q}(n\tau_t)$; then, the EE optimization problem can be rewritten as

$$
(\mathrm{P}\,2)\colon \max_{\mathrm{q}[l]} \frac{\bar{R}_{total}(\mathrm{q}[l])}{\bar{E}(\mathrm{q}[l])} = \frac{B \sum_{l=1}^{L} \sum_{k=1}^{K} \sum_{n=1}^{N} \log_2 \left( 1 + \dfrac{p_{k,n}}{\sum_{i=n+1}^{N} p_{k,i} + \alpha(H^2 + \|\mathrm{q}[l] - \mathrm{W}_{k,n}\|^2)} \right)}{\sum_{l=1}^{L} \left( c_1 \|\mathrm{v}[l]\|^3 + \dfrac{c_2}{\|\mathrm{v}[l]\|} \right)}
\tag{13}
$$

s.t. $\mathrm{C}\,1 : \bar{R}_{total}(\mathrm{q}[l]) \geq R_{target}\ , \forall l$

$\mathrm{C}\,2 : p_{k,n} < P_{\max}$

$\mathrm{C}\,3 : \sum_{i=1}^{N} p_{k,i} \leq P_{\max}\ .$

## 4 Problem transformation and algorithm design

In this section, we first propose three user clustering algorithms to optimize the transmission of data between ground users and UAVs. In particular, the basic principle of these three

algorithms is to cluster users according to the size of the channel conditions. To facilitate calculation and validation, the total number of ground users is set as $U$. Then, the trajectory and resource allocation of the UAV are optimized through alternate iterations to achieve the objective of maximizing the EE based on the clustering algorithms. An exhaustive search algorithm is then used to obtain the horizontal coordinates of the UAV.

## 4.1 User clustering in downlink NOMA

In various NOMA scenarios, ground users are divided into multiple clusters, and each cluster contains a certain number of users in the process of sending signals. Each subcarrier shares a cluster resource. To enhance spectral efficiency, this work expands the differences in channel conditions among cluster users when designing the user clustering algorithm. The sum rate and EE are considered performance indicators in the proposed algorithms.

**4.1.1 Greedy clustering algorithm.** The greedy clustering algorithm is a classic matching algorithm that considers only the simplest case of two users per cluster. For maximum channel gain differences in in-cluster users, we assume that a high-channel quality indicator (CQI) is combined with a low CQI for users as much as possible [28]. The main principle of the greedy clustering algorithm is ordering users from low to high according to the CQI index first and then matching the sorted users in order of initial and final information so that the differences in channel gains are tremendous among in-cluster users. The specific process is shown in Algorithm 1, which always selects two users who have the largest channel gain differences for clustering from the current and remaining users until all users have been clustered. Mathematically, this means that the k- th sorted user is matched with the (U-k+1)$th$ sorted user and constitutes a cluster conjointly.

**Algorithm 1 Greedy clustering algorithm**

1. **Sorted:** Users are sorted by increasing channel gain:
$$0 < \|h_1(t)\|^2 \le \|h_2(t)\|^2 \le \cdots \le \|h_{\frac{U}{2}}(t)\|^2 \le \cdots \le \|h_U(t)\|^2 .$$

2. **Input:** Users are divided into $K$ cluster, and each user cluster contains $N$ users.

3. **Clustering:** Group users into clusters for downlink NOMA:
   $1st$ cluster $A_1 = \{\|h_1(t)\|^2, \|h_U(t)\|^2\}$,
   $2nd$ cluster $A_2 = \{\|h_2(t)\|^2, \|h_{U-1}(t)\|^2\}$,
   ...
   $kth$ cluster $A_k = \{\|h_k(t)\|^2, \|h_{U-k+1}(t)\|^2\}$,
   ...
   $Kth$ cluster $A_K = \{\|h_{\frac{U}{2}}(t)\|^2, \|h_{\frac{U}{2}+1}(t)\|^2\}, K = \frac{U}{2}$.

4. **Cluster size:** if ((U mod N)==0)
   then uniform cluster size else different cluster size.

For ease of exposition, we display user clustering for $N$-user NOMA clusters in the downlink, where the total number of users $U$ is set to 12 and $N$ is set to 2 to avoid excessive SIC complexity. Sort users $U_1, U_2, \ldots, U_{12}$ from left to right according to their channel gains in Fig 2.

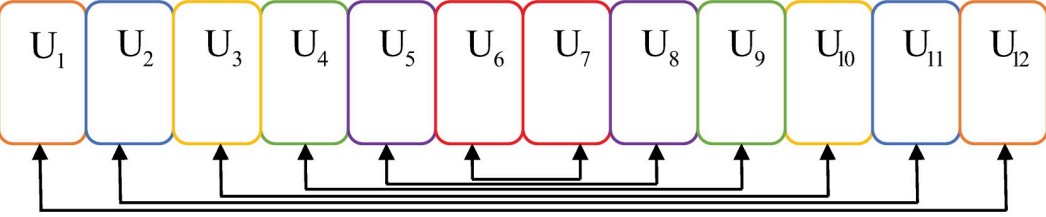

**Fig 2. Cluster results of the greedy clustering algorithm.**

First, users $U_1$ and $U_{12}$, who have the most different channel gains, are bound to a cluster. Then, users $U_2$ and $U_{11}$ become the second cluster of residual users. By that analogy, the clustering result is shown in Fig 2. In particular, rectangular blocks with the same color represent the same cluster in this work. However, a trade-off needs to be considered when clustering users because the algorithm neglects middle interuser differences in channel gains. Therefore, the greedy clustering algorithm is a local optimal algorithm considering partial users with the maximum differences in channel gain rather than a global optimal solution.

**4.1.2 Suboptimal clustering algorithm.** Owing to the need to solve the existing problem of greedy matching algorithms, a suboptimal clustering algorithm is further proposed by utilizing uniform channel gain differences. The algorithm emphasizes maintaining relative average differences between in-cluster users of all clusters. Overall, users are classified into class A and class B on the basis of channel gains, followed by intergroup clustering in such a way that middle users are primely accommodated [27]. The number of users in class A is denoted as $M$, which is provided with a much smaller channel gain than that of users in class B; this number is denoted as U- M.

**Algorithm 2 Suboptimal clustering algorithm**

1. **Sorted:** Users are sorted by increasing channel gain:
   $0 < \|h_1(t)\|^2 \leq \|h_2(t)\|^2 \leq \cdots \leq \|h_M(t)\|^2 \ll \|h_{M+1}(t)\|^2 \leq \cdots \leq \|h_U(t)\|^2$.
2. **Input:** Determine the total number of users in class A and class B.
   The number of cluster $K$: if $M < \frac{U}{2}$, $K = M$; else if $M \geq \frac{U}{2}$, $K = \frac{U}{2}$.
3. **Clustering:** Group users into clusters for downlink NOMA:
   $1st$ cluster $B_1 = \{\|h_1(t)\|^2, \|h_{K+1}(t)\|^2, \|h_{2K+1}(t)\|^2, \cdots, \|h_{U-K-1}(t)\|^2\}$,
   $2nd$ cluster $B_2 = \{\|h_2(t)\|^2, \|h_{K+2}(t)\|^2, \|h_{2K+2}(t)\|^2, \cdots, \|h_{U-K-2}(t)\|^2\}$,
   ...
   $kth$ cluster $B_k = \{\|h_k(t)\|^2, \|h_{K+k}(t)\|^2, \|h_{2K+k}(t)\|^2, \cdots, \|h_{U-K-k}(t)\|^2\}$,
   ...
   $Kth$ cluster $B_K = \{\|h_K(t)\|^2, \|h_{2K}(t)\|^2, \|h_{3K}(t)\|^2, \cdots, \|h_U(t)\|^2\}$.
4. **Cluster size:** if ((U mod N)==0)
   then uniform cluster size, else different cluster size.

The suboptimal clustering algorithm is shown in Algorithm 2, which considers an ordinary case in which a large number of users are equably distributed. Step 3 demonstrates the possibility of multiple users grouping in clusters for downlink NOMA. This means that the minimum channel gain of a user in class A is clustered with the minimum of the other user in class B, followed by clustering the next minimum users of both classes and so on. This process will stop when the maximum channel gain of the user from both classes is clustered by analogy. In brief, the outstanding advantage of the suboptimal clustering algorithm is that it accommodates middle users with channel gain.

After utilizing Algorithm 2, the scenario exhibited in Fig 2 can be replanned in Figs 3 and 4. The average channel gain differences among in-cluster users follow a relatively consistent clustering behavior. The cases refer to two users and three users in the cluster in Figs 3 and 4, respectively. The users are divided into class A and class B, and each cluster requests users with

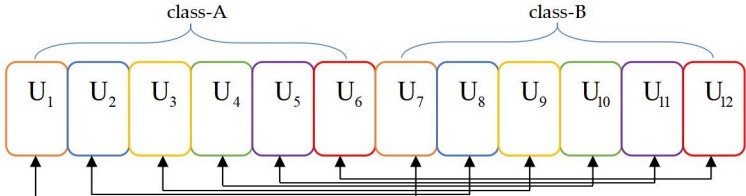

**Fig 3. Cluster results of the suboptimal clustering algorithm when $M$ is 6.**

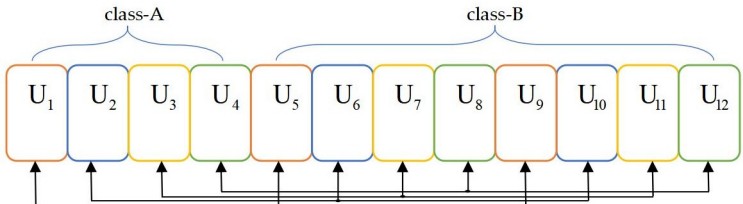

**Fig 4. Cluster results of the suboptimal clustering algorithm when** $M$ **is 4.**

the same number and color. When the number of users in a cluster increases, the corresponding scheme can more suitably cluster users based on correspondence between user clusters.

**4.1.3 Hybrid clustering algorithm.** When users are ranked by increasing channel gain, the difference in the CQI index between adjacent user clusters becomes increasingly smaller if a suboptimal clustering algorithm is utilized for clustering, which makes it difficult to carry out subsequent operations such as signal separation [10]. Hence, a hybrid clustering algorithm is proposed based on the previous two algorithms. This hybrid clustering algorithm exploits the greedy clustering algorithm to cluster the users in two terminals and uses the suboptimal clustering algorithm to cluster the users in the middle. This scheme realizes clustering of users with high CQI and low CQI while minimizing intermediate user issues by making trade-offs [28].

**Algorithm 3 Hybrid clustering algorithm**

```
1. Sorted: Users are sorted by increasing channel gain:
   0 < ‖h₁(t)‖² ≤ ‖h₂(t)‖² ≤ ⋯ ≤ ‖h_U(t)‖²
2. Input: Determine the number of user clusters K, and tp is set as:
   tp = ⌊U/2 - U/4⌋.
3. Clustering: Group users into clusters for downlink NOMA:
   If 1 ≤ k ≤ tp, kth cluster: C_k = {‖h_k(t)‖², ‖h_{U+1-k}(t)‖²};
   else if tp < k ≤ U/2, kth cluster: C_k = {‖h_k(t)‖², ‖h_{U/2+k-tp+1}(t)‖²}.
4. Cluster size: if ((U mod N)==0)
   then uniform cluster size, else different cluster size.
```

A threshold $tp$ that distinguishes intermediate from terminal users is designed according to the principle of the hybrid clustering algorithm in Algorithm 3. Based on the greedy clustering algorithm, $\|h_1(t)\|^2$ and $\|h_U(t)\|^2$ are the lowest and highest CQI users, respectively, after being clustered. $\|h_2(t)\|^2$ and $\|h_{U-1}(t)\|^2$ are the second-lowest and second-highest CQI users, respectively, after being clustered in the remainder. By parity of manner, it is deduced that the $kth$ cluster includes $\|h_k(t)\|^2$ and $\|h_{U+1-k}(t)\|^2$ when $1 \leq k \leq tp$. Once the threshold $tp$ is reached, the clustering scheme with uniform channel gain differences from this point to clusters is employed. This means that $\|h_k(t)\|^2$ and $\|h_{\frac{U}{2}+k-tp+1}(t)\|^2$ constitute the $kth$ cluster when $tp < k \leq \frac{U}{2}$. The scenario presented in Figs 2 and 3 can be expressed in Fig 5 after hybrid clustering is implemented.

Fig 5 shows the clustering situation, where the total number of sorted users U is 12 and $tp$ is 4. The $U_4$ in class-A and $U_9$ in class-B makes an accommodation to the surrounding users $U_5$ and $U_6$, $U_7$ and $U_8$, respectively. As a result, $U_4$ is clustered with $U_9$, $U_5$ is clustered with $U_8$, and $U_6$ is clustered with $U_7$ in a manner consistent with the difference in channel gain. This means that if λ users are equipped with subequal gains to each other and fail to be grouped, the $\lfloor \frac{\lambda}{2} \rfloor$ users with both adjacencies will accommodate them utilizing a suboptimal clustering algorithm. Due to the combination of these two algorithms, the hybrid clustering algorithm avoids the situation in which the channel gain differences are decreased in some user clusters.

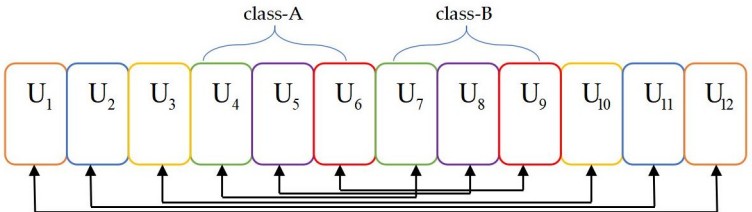

**Fig 5. Cluster results of the hybrid clustering algorithm.**

## 4.2 Maximum EE problem

As shown in Section 3, the nonconvex optimization problem (P1) is transformed into an approximately equivalent convex optimization problem (P2) via the bisection method and Dinkelbach method [31]. Problem (P2) is deemed a fractional maximization problem with a convex numerator and a convex denominator, as well as all convex constraints. Thus, the solutions of the EE problem are computed with polynomial complexity.

**4.2.1 Problem transformation.**   *Theorem 1:* Let $\eta^*$ denote the optimal EE, and let $p^*$ and q $[l]^*$ denote the optimal solution of problem (P2).

$$\eta^* = \max \frac{\bar{R}_{total}(\mathrm{p}, \mathrm{q}[l])}{\bar{E}(\mathrm{p}, \mathrm{q}[l])} \tag{14}$$

if and only if,

$$\max \left\{ M(\mathrm{p}, \mathrm{q}[l], \eta) = \bar{R}_{total}(\mathrm{p}, \mathrm{q}[l]) - \eta^* \bar{E}(\mathrm{p}, \mathrm{q}[l]) \right\}$$
$$= \bar{R}_{total}(\mathrm{p}^*, \mathrm{q}[l]^*) - \eta^* \bar{E}(\mathrm{p}^*, \mathrm{q}[l]^*) = 0. \tag{15}$$

*Proof of Theorem 1.* From Eq (14), we have equation $\max\limits_{\mathrm{p}^*, \mathrm{q}[l]^*} \bar{R}^*_{total}(\mathrm{p}^*, \mathrm{q}[l]^*) - \eta \bar{E}^*(\mathrm{p}^*, \mathrm{q}[l]^*)$ based on the Dinkelbach method. If $Q^*(\mathrm{p}^*, \mathrm{q}[l]^*)$ is used to denote $\bar{R}^*_{total}(\mathrm{p}^*, \mathrm{q}[l]^*) - \eta^* \bar{E}^*(\mathrm{p}^*, \mathrm{q}[l]^*)$, we can obtain $\eta = \eta^* \Leftrightarrow M(\mathrm{p}, \mathrm{q}[l], \eta) = Q^*(\mathrm{p}^*, \mathrm{q}[l]^*, \eta^*) = 0$, which has been proven in the formal literature [31]. $M(\mathrm{p}, \mathrm{q}[l], \eta)$ denotes the optimal value of $\max\limits_{\mathrm{p}^*, \mathrm{q}[l]^*} Q^*(\mathrm{p}^*, \mathrm{q}[l]^*)$. For $\eta$, $M(\mathrm{p}, \mathrm{q}[l], \eta)$ is a monotonically increasing convex function. Thus, problem (P2) is written as Eq (16).

Our objective is to update $\eta$ in each iteration until it converges to $\eta^*$ or reaches the maximum number of iterations $O_{\max}$. We assume that the iteration index is $c$ and that the optimization problem for a given parameter in each iteration is $\eta^c$. Thus, there is

$$\max_{\mathrm{p}, \mathrm{q}[l]} \left\{ M(\mathrm{p}, \mathrm{q}[l], \eta^c) = \bar{R}_{total}(\mathrm{p}, \mathrm{q}[l]) - \eta^c \bar{E}(\mathrm{p}, \mathrm{q}[l]) \right\}$$

$$\text{s.t. } C1 : \bar{R}_{total}(\mathrm{q}[l]) \geq R_{target} \text{ , } \forall l$$

$$C2 : p_{k,n} < P_{\max}$$

$$C3 : \sum_{i=1}^{N} p_{k,i} \leq P_{\max} \tag{16}$$

$$C4 : p_{k,j} < \frac{P_t}{2^{N-1}}.$$

After using the user clustering algorithm, the restriction (C4) is added, which guarantees that the user with the maximum transmission power allocation to the highest channel gain in the NOMA cluster must be smaller than $\frac{P_t}{2^{N-1}}$ [27]. $P_t$ is expressed as the transmission power budget of the cluster. The specific process of the Maximum EE algorithm is shown in Algorithm 4.

**Algorithm 4 Maximum EE algorithm**

```
1: Initialization: Let iteration index o = 1 and maximum iteration
      number is O_max. Set η¹ = 0, error tolerance δ ≪ 1.
2: repeat
3:    For a given η° to solve the equivalent problem (16) and obtain the
      optimal solution p° and q[l]°.
4:    Let η° = max   R̄_total(p,q[l])    .
             p,q[l]    Ē(p,q[l])
5:    o = o + 1.
6: until the iteration index o > O_max or |(M(p, q[l], η)| ≤ δ.
7: Let p* = p^{o-1}, q[l]* = q[l]^{o-1} and η* = η^{o-1}.
```

**4.2.2 Power allocation.**   The fixed q[l] is taken into consideration in this segment. Hence, to facilitate calculation, the objective function in (16) is restated as

$$\Psi(\mathrm{p}) = f(\mathrm{p}) - g(\mathrm{p}), \tag{17}$$

where

$$f(\mathrm{p}) = B \sum_{l=1}^{L} \sum_{k=1}^{K} \sum_{n=1}^{N} \log_2 \left( \alpha(H^2 + \|\mathrm{q}[l] - \mathrm{W}_{k,n}\|^2) + \sum_{n=1}^{N} p_{k,n} \right) - \eta^c \sum_{l=1}^{L} \left( c_1 \|\mathrm{v}[l]\|^3 + \frac{c_2}{\|\mathrm{v}[l]\|} \right), \tag{18}$$

$$g(\mathrm{p}) = B \sum_{l=1}^{L} \sum_{k=1}^{K} \sum_{n=1}^{N} \log_2 \left( \alpha(H^2 + \|\mathrm{q}[l] - \mathrm{W}_{k,n}\|^2) + \sum_{i=n+1}^{N} p_{k,i} \right). \tag{19}$$

Both $f(\mathrm{p})$ and $g(\mathrm{p})$ are concave functions. The function $g(\mathrm{p})$ can be approximated as $g(\mathrm{p}^I) + \nabla g(\mathrm{p}^I)(\mathrm{p} - \mathrm{p}^I)$ by its first-order Taylor expansion [32]. $\nabla g(\mathrm{p}^I)$ denotes the gradient of $g(\mathrm{p})$ at $\mathrm{p}^I$, which is calculated as

$$\nabla g(\mathrm{p}^I) = B \sum_{l=1}^{L} \sum_{k=1}^{K} \sum_{n=1}^{N} \frac{1}{(\alpha(H^2 + \|\mathrm{q}[l] - \mathrm{W}_{k,n}\|^2) + \sum_{i=n+1}^{N} p_{k,i}) \ln 2}. \tag{20}$$

Therefore, problem (17) is further transformed to

$$\max_{\mathrm{p}} \left\{ \Psi(\mathrm{p}) = f(\mathrm{p}) - g(\mathrm{p}^I) - \nabla g(\mathrm{p}^I)(\mathrm{p} - \mathrm{p}^I) \right\}$$

$$\text{s.t. } C1 : \bar{R}_{total}(\mathrm{q}[l]) \geq R_{target}$$

$$C2 : p_{k,n} < P_{\max}$$

$$C3 : \sum_{i=1}^{N} p_{k,i} \leq P_{\max}$$

$$C4 : p_{k,j} < \frac{P_t}{2^{N-1}}. \tag{21}$$

Problem (21) is a standard convex optimization problem that can be effectively solved by CVX, a convex program solver. The proposed power allocation algorithm is presented in Algorithm 5. In each iteration, the solution $\mathrm{p}^{I+1}$ is generated as the optimal solution of (21) at the last iteration. From $f(\mathrm{p}^I) - g(\mathrm{p}^I) \leq f(\mathrm{p}^{I+1}) - g(\mathrm{p}^I) - \nabla g(\mathrm{p}^I)(\mathrm{p}^{I+1} - \mathrm{p}^I)$, we obtain $g(\mathrm{p}) \leq g(\mathrm{p}^I) - \nabla g(\mathrm{p}^I)(\mathrm{p} - \mathrm{p}^I)$ at any p. Then, we derive that $f(\mathrm{p}^I) - g(\mathrm{p}^I) \leq f(\mathrm{p}^{I+1}) - g(\mathrm{p}^{I+1})$. Thus, the objective value $\Psi(\mathrm{p})$ is improved with each iteration, and the power allocation sequence $\{\mathrm{p}^I\}$ converges by the Cauchy theorem. Moreover, the proposed algorithm is combined with three types of user clustering algorithms and two iterative algorithms. According to the total number of

**Table 3. The table about parameter settings for scenario.**

| Parameter | Value | Implication |
|-----------|-------|-------------|
| $U$ | 24 | number of total users |
| $K$ | 12 | number of user clusters |
| $N$ | 2 | number of user in each cluster |
| $B$ | $1 MHz$ | channel bandwidth |
| $\sigma^2$ | -110dBm | the true noise variance |
| $\delta$ | $1 \times 10^{-8}$ | error tolerance |
| $t$ | 1s,2s | flight time of UAV |
| $\beta_0$ | -50dB | the channel power gain at the reference distance $d_0 = 1m$ |
| $c_1$ | $9.26 \times 10^{-4}$ | parameters related to the UAV |
| $c_2$ | 2250 | parameters related to the UAV |
| $G_0, G_1$ | 1 | index of directional antenna gain |

users $U$, the computational complexity of the greedy clustering algorithm is denoted as $O(U^3)$, which consists of the complexity of bubble sorting $O(U^2)$ and the complexity of searching for clusters $O(U)$. The complexities of the suboptimal clustering algorithm and hybrid clustering algorithm are set as $O(U^2 log_2 U)$ and $O(U^3 log_2 \frac{U}{2})$, respectively. In addition, the computational

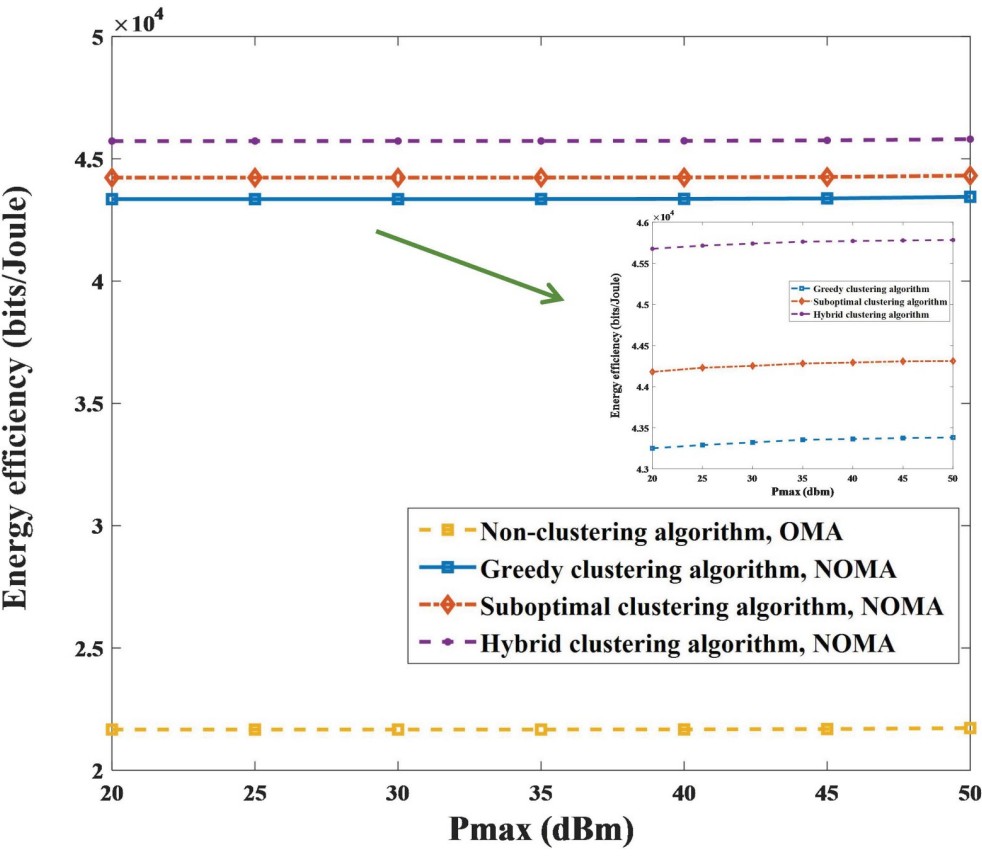

**Fig 6. Relationship between EE and $P_{\max}$.**

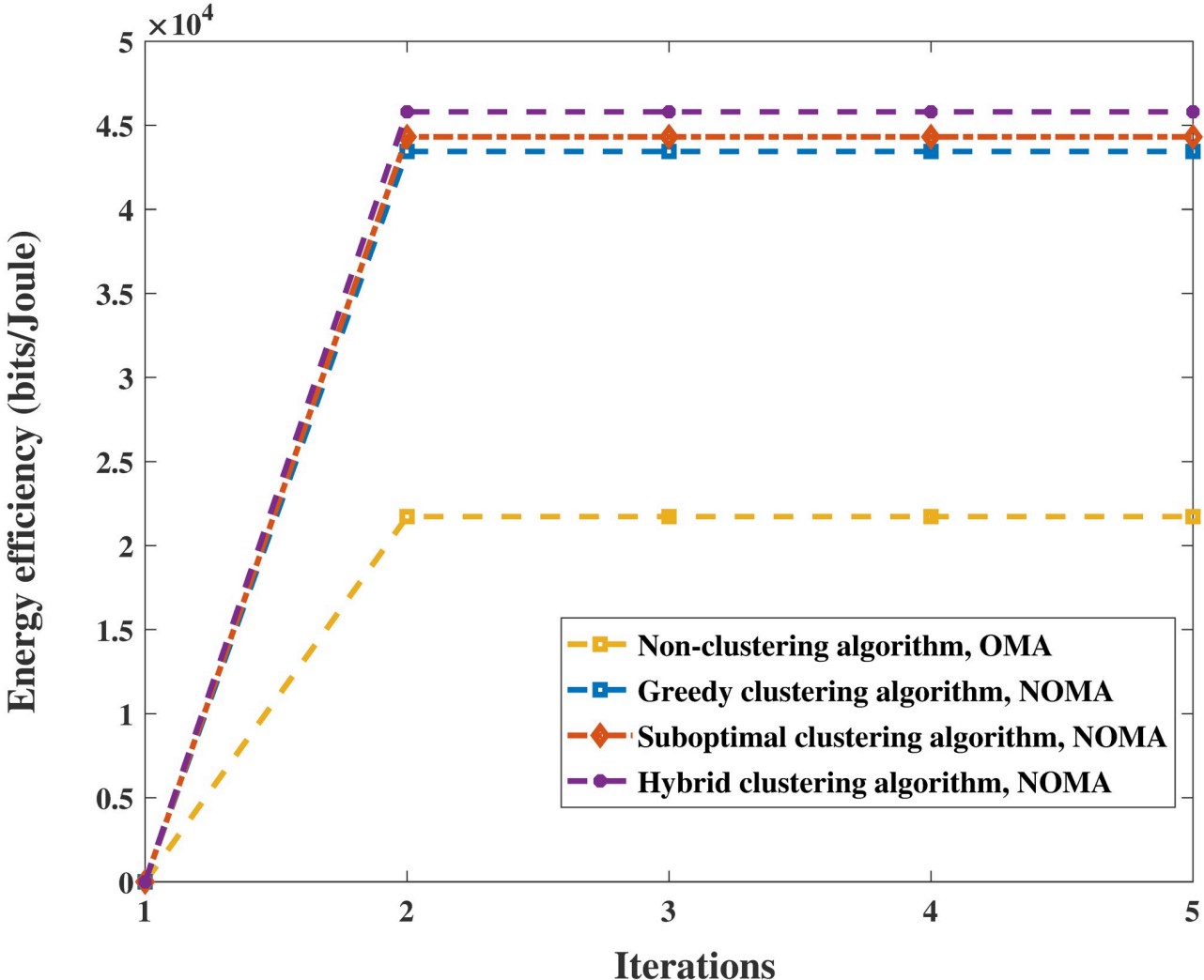

**Fig 7. Relationship between EE and iteration in $P_{max} = 50 dBm$.**

complexity of the two iterative algorithms is denoted as $O(OE)$, where $O$ is the iteration number of Algorithm 4 and bounded by $O_{max}$, and $E = IO(U^3)$ denotes the computational complexity of Algorithm 5 since the complexity of (21) is $O(U^3)$ and $I$ is the iteration number bounded by $I_{max}$.

**Algorithm 5 Power allocation algorithm**

```
 1: Initialization: Let iteration index I = 0and maximum iteration
    number is Imax, error tolerance ε ≪ 1.
 2: repeat
 3:   For a given η^o and p^I to solve the equivalent problem (19) and
      obtain the optimal solution p^opt.
 4:   Set p^I = p^opt and calculate Ψ(p^I) = f(p^I) - g(p^I).
 5:   I = I + 1.
 6: until the iteration index I > Imax or |Ψ(p^I) - Ψ(p^{I-1})| ≤ ε.
 7: Let p^o = p^I.
```

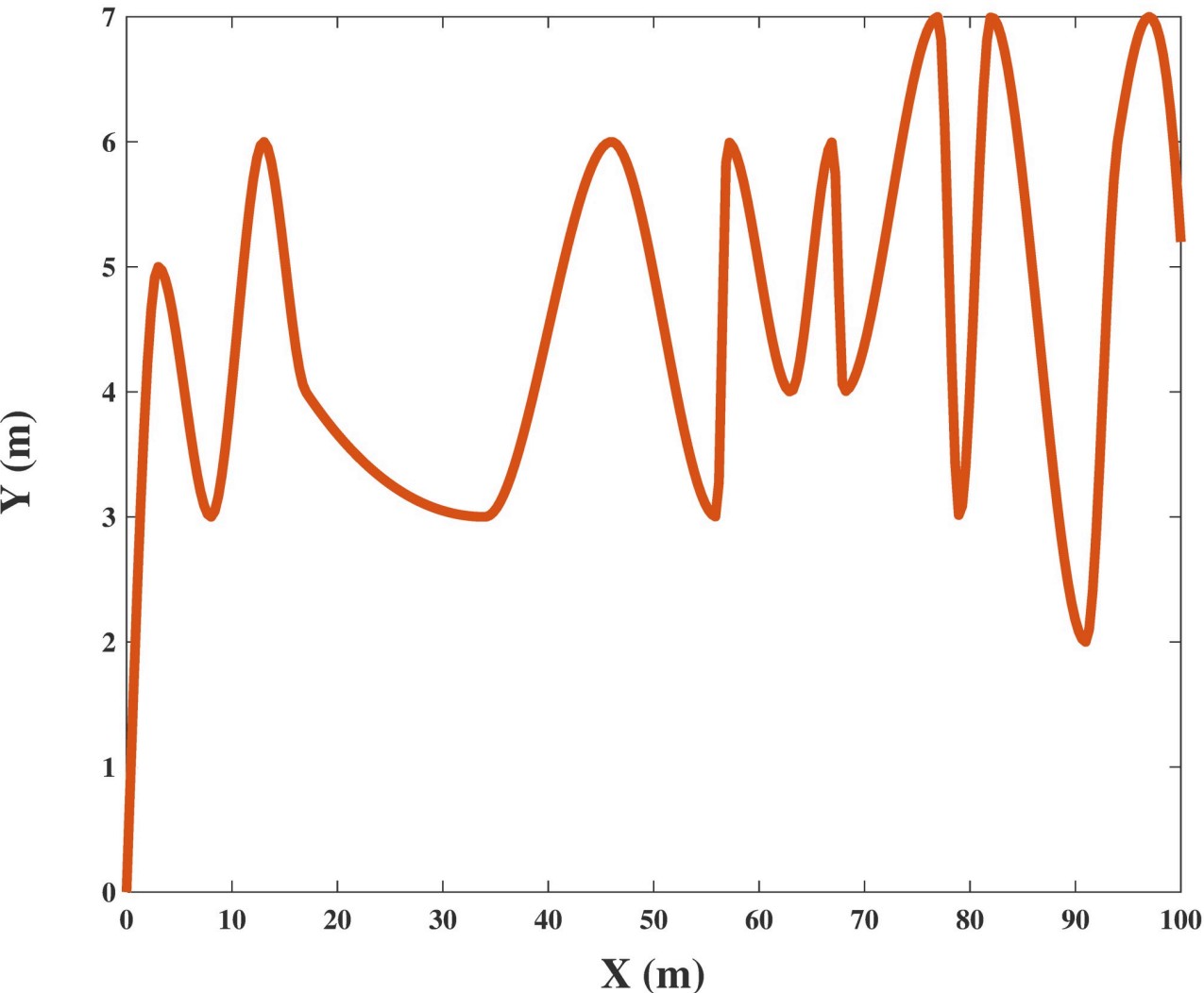

**Fig 8. The simulation result with UAV trajectory in horizontal position when height = 100m.**

## 5 Numerical results

In this section, simulation results are utilized to evaluate the effectiveness of the proposed algorithm. All ground users are randomly distributed in a suburban environment with a range of [±200, ±200]. A single UAV is used as an ABS to transmit information to users. First, three user clustering algorithms are investigated by altering the numerical value of $P_{max}$ to verify that the performance of the hybrid clustering algorithm is superior to that of the other algorithms. Then, the optimal horizontal position of the UAV is obtained by utilizing an exhaustive search method based on a hybrid clustering algorithm with multiple alternating iterations. Finally, the influence of UAV height on EE is explored on the basis of previous works. The parameter settings are given in Table 3.

A comparison of the proposed algorithms with the conventional OMA method under different parameters is shown in Figs 6 and 7. Fig 6 shows the superiority of the power allocation scheme in Algorithm 5 in combination with user clustering algorithms and UAV trajectory

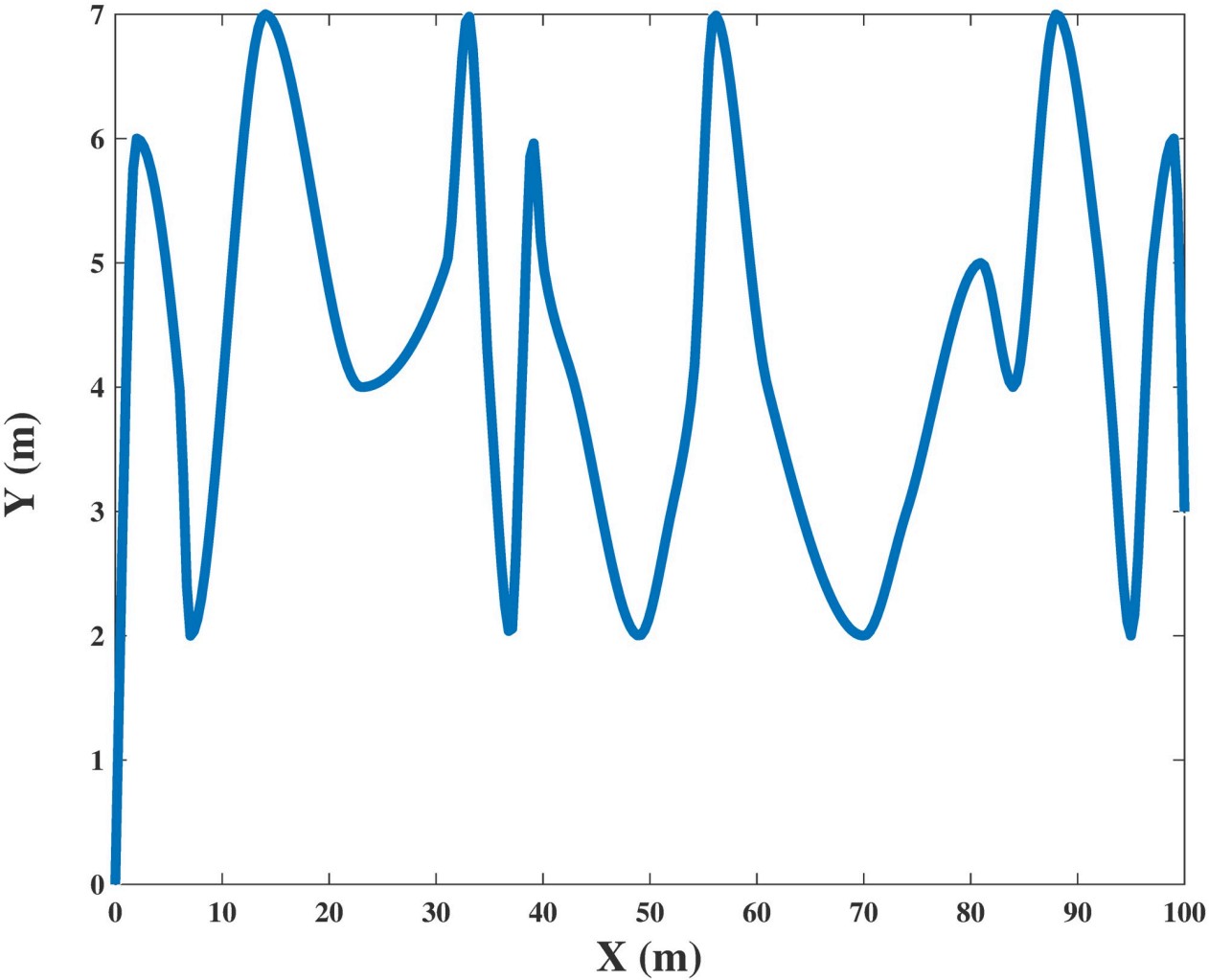

**Fig 9. The simulation result with UAV trajectory in horizontal position when height = 200m.**

optimization. On the one hand, the EE of the system increases with $P_{max}$ and gradually converges after $P_{max} = 35dBm$ with the same user clustering algorithm. The simulation results show that the performance of the EE algorithm when clustering algorithms are used with NOMA is better than that of the nonclustering algorithms with OMA. On the other hand, the increase in EE with the hybrid clustering algorithm is superior to that of the other two algorithms, as shown in Figs 6 and 7. It is reasonable that the hybrid clustering algorithm considers and corrects the shortcomings of the greedy clustering algorithm and suboptimal clustering algorithm. In addition, as shown in Fig 7, the variation in EE with alternating iterations is explored. After one or two iterations, the EE of Algorithm 4 converges to the approximately global optimizal solution by calculating the space convergence feature and optimality of the proposed algorithm. Clearly, the superiority of the NOMA scheme is revealed in that it produces greater performance gain when users are equipped with highly different channel conditions. The simulations indicate that the performance of EE with the proposed algorithms is improved by more than 99.6%, 104%, and 111%, respectively.

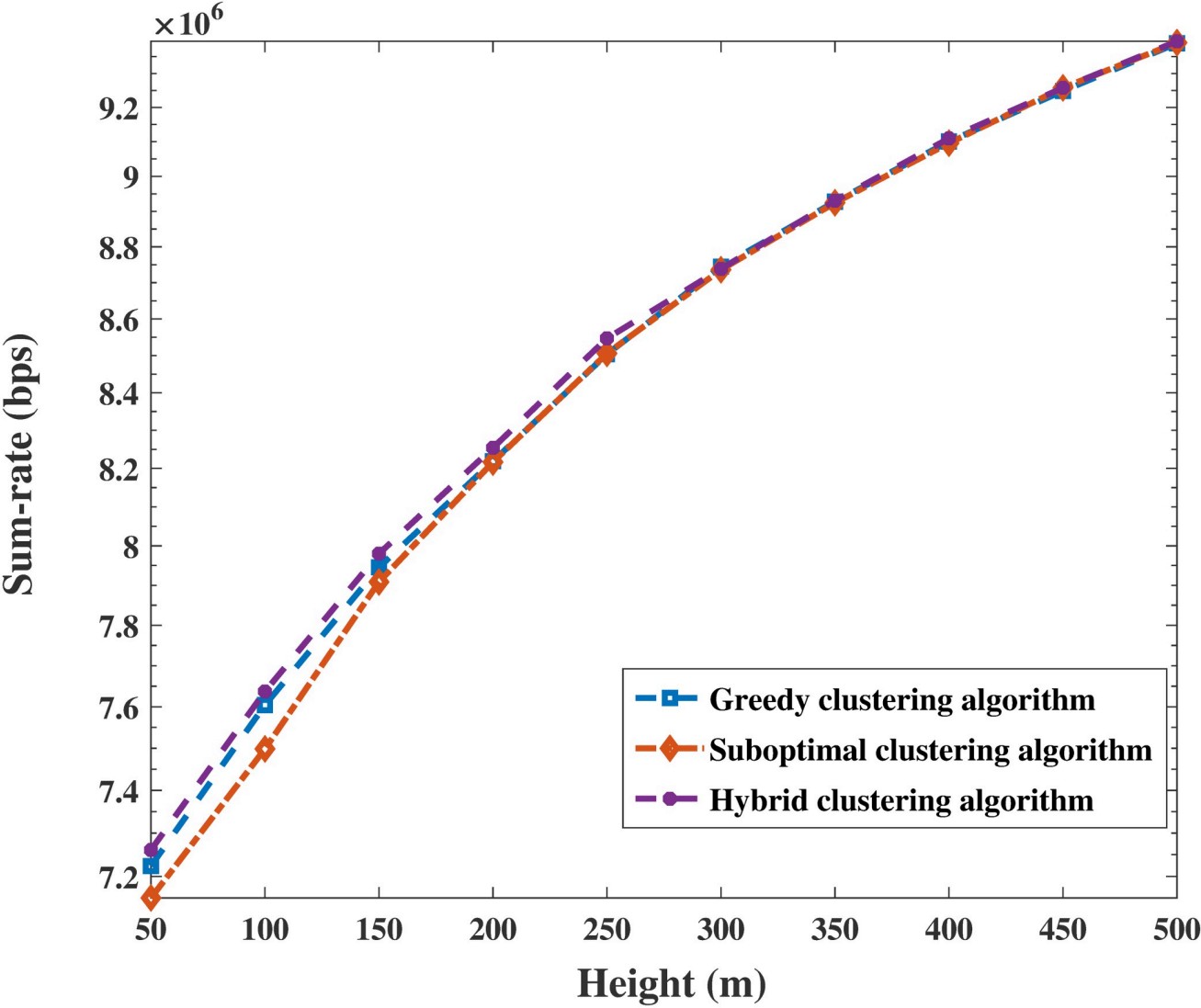

**Fig 10. Relationship between sum rate and height.**

The horizontal flight trajectory of the UAV obtained by Algorithm 4 based on the hybrid clustering algorithm for a given maximum transmission power is shown in Figs 8 and 9. The height of the UAV is fixed at 100 m and 200 m, respectively. Due to the intractable nature of the problem, multiple iterations based on Algorithm 4 are utilized in the process of simulation. Advanced EE performance can be achieved by carefully considering the power allocation and properly designing the trajectory of the UAV. The results show that the UAV obtains the optimal EE when flying approximately straight with velocity $v = 30m/s$ in the designed scene, which produces a great signal transmission effect from the UAV to ground user clusters. In addition, the flight velocity of the UAV in each time slot can be calculated from the derivative of its location $q(t)$. UAVs become more flexible with increasing flight cycle $T$. Thus, with our proposed scheme, the mobility of UAVs can be more

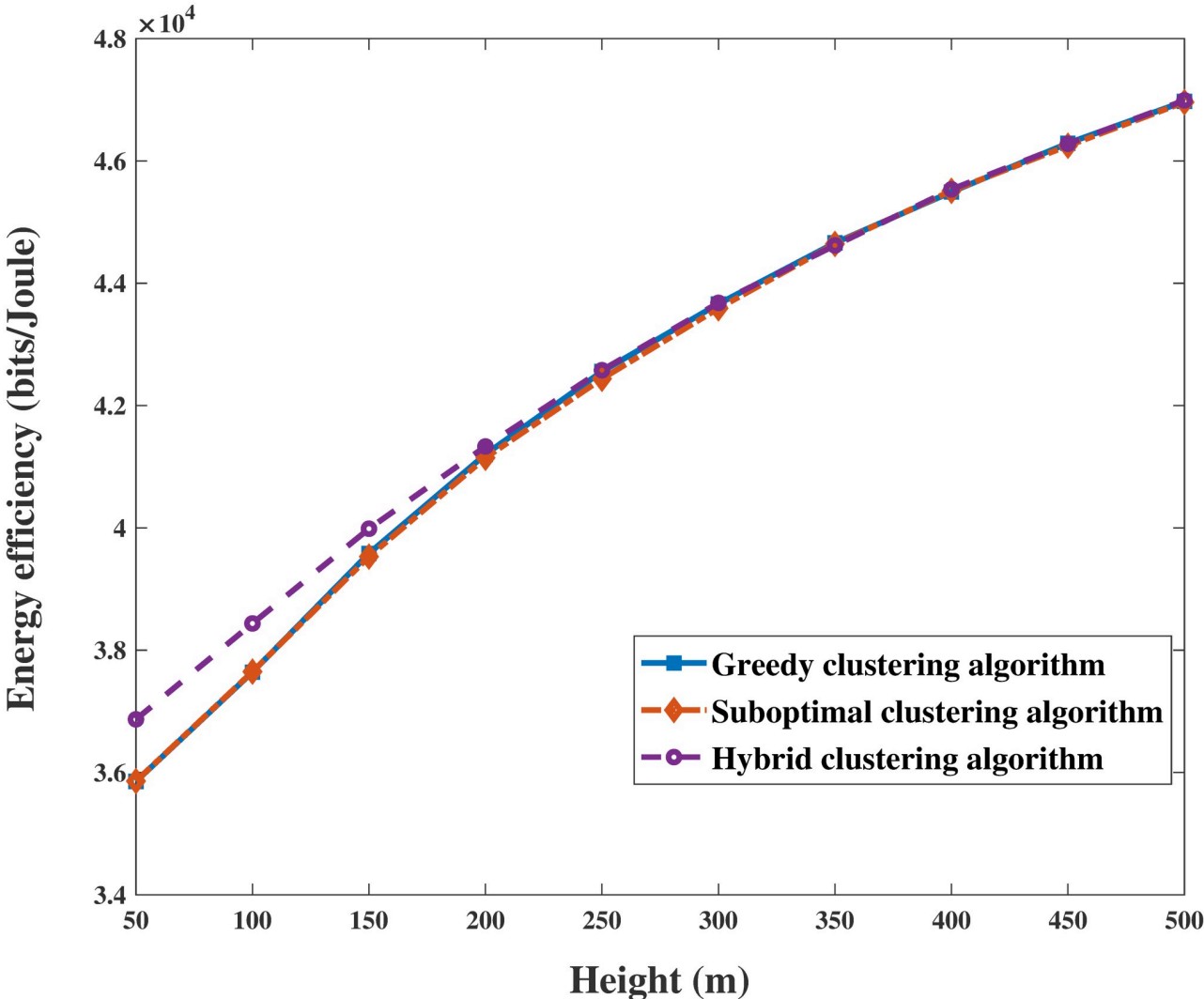

**Fig 11. Relationship between EE and height.**

efficiently utilized to improve the performance of EE. Nevertheless, there are obvious drawbacks to this scheme, such as high complexity and the large amount of time required for simulation operations.

As shown in Figs 10 and 11, the results clearly demonstrate the convergence of the sum rate and EE of the system versus the height of the UAV for identical parameters. The sum rate and EE increase with increasing height. Then, both objectives become saturated and reach a tradeoff when the UAV's height is sufficiently large. This trend occurs because the energy sustaining flight is multiplied, and the circuit power consumption of antennas becomes a primary factor in the system performance. In addition, Figs 12 and 13 are enlarged images of Fig 11 that display the EE compared with that of different clustering algorithms for heights = 200 m and 400 m. In addition, the performances of the three clustering algorithms gradually increase with increasing height, as shown in Figs 10 and 11. It is reasonable that the greater the signal

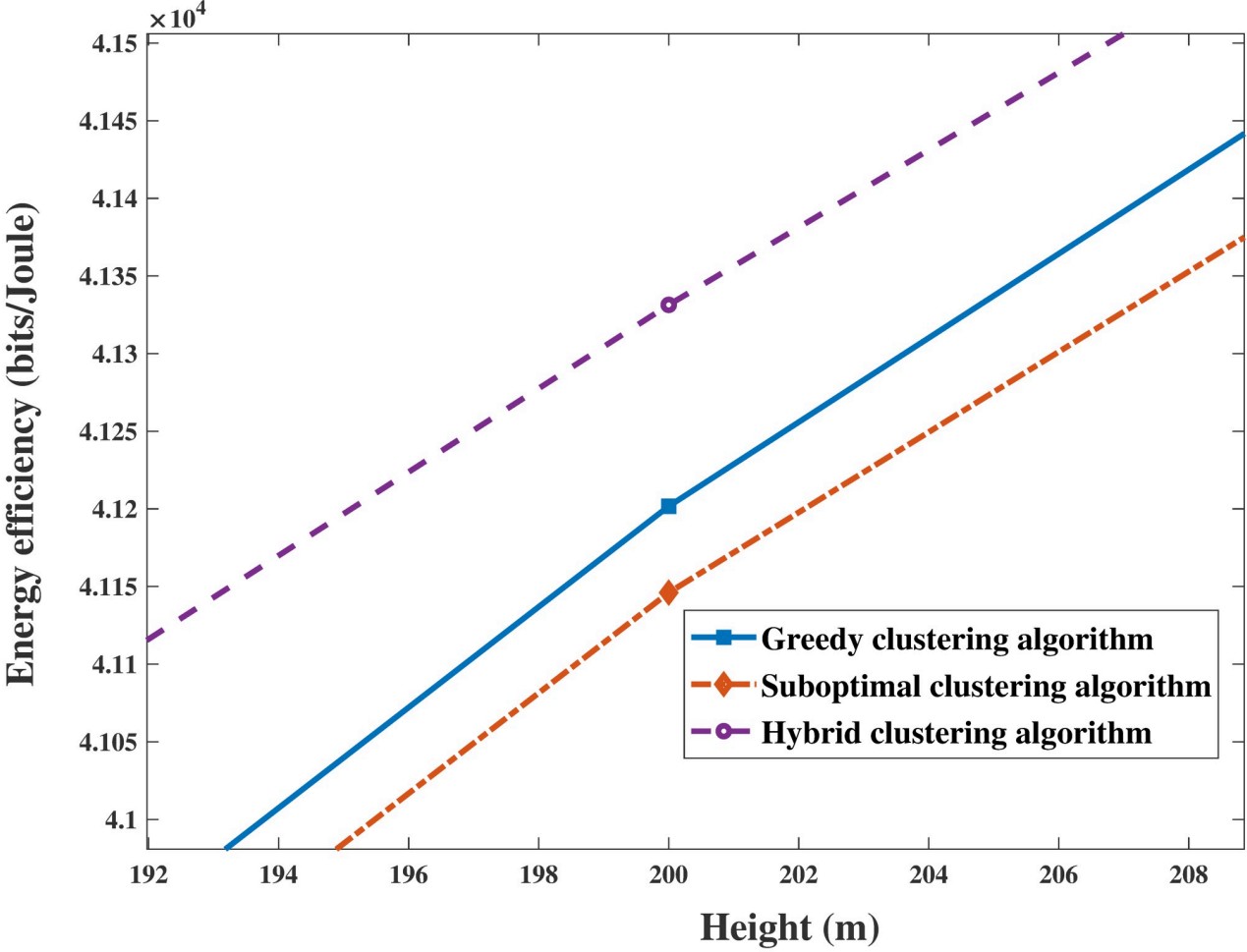

**Fig 12. The enlarged drawing with Fig 11 in height = 200m.**

transmission distance between a UAV and ground users is, the less noticeable the performance of clustering user algorithms during flying.

## 6 Conclusions

In this work, we propose an energy-efficient model of UAV-related NOMA and reconstruct the EE problem by transforming the nonconvex problem into an equivalent convex optimization problem. Accordingly, an EE scheme is designed to jointly optimize UAV resource allocation and trajectory planning under the constraints of velocity and mobility in the downlink. First, three user clustering algorithms are applied to cluster ground users. Then, an iterative optimization algorithm is proposed. The simulation results indicate that the performance of the EE has improved by 99.6%, 104%, and 111%, respectively. Nevertheless, the situation with multiple UAVs is neglected, and the priority of user requirements for information lack consideration; these factors will be explored in future work.

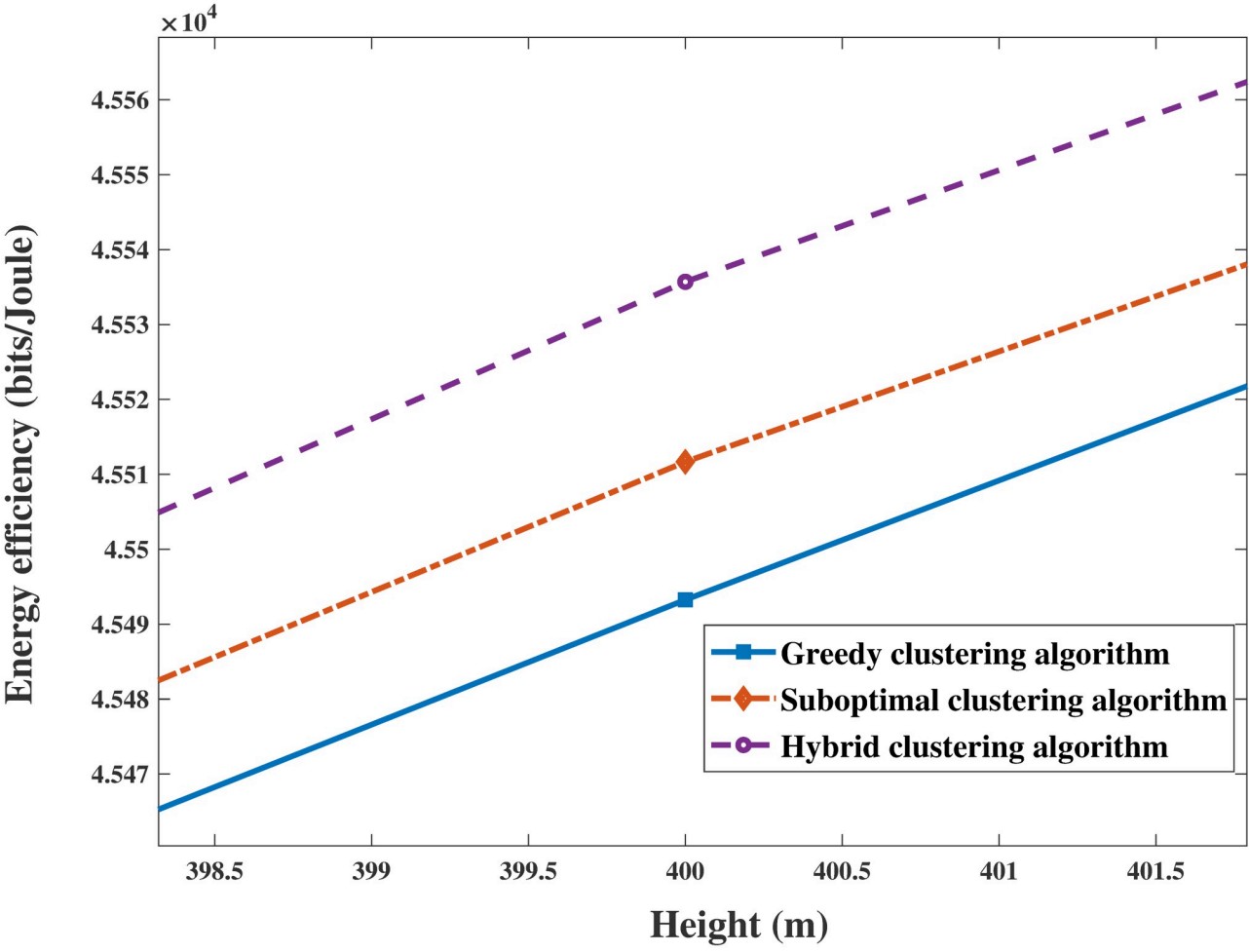

**Fig 13. The enlarged drawing with Fig 11 in height = 400m.**

## Supporting information

**S1 Data.**
(ZIP)

## Acknowledgments

Thanks for helps of Hebei Normal University and Hebei Provincial Key Laboratory of Information Fusion and Intelligent Control.

## Author Contributions

**Conceptualization:** Yucong Zhou.

**Methodology:** Xiaozi Jin, Shuang Zhang.

**Writing – original draft:** Yucong Zhou.

**Writing – review & editing:** Huilong Jin, Shuang Zhang.

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
