## [Decision Letter · Decision Letter 0]

13 Aug 2023

PONE-D-23-21957Energy-Efficient UAV Communication: A NOMA Scheme with Resource Allocation and Trajectory OptimizationPLOS ONE

Dear Dr. Zhang,

Thank you for submitting your manuscript to PLOS ONE. After careful consideration, we feel that it has merit but does not fully meet PLOS ONE’s publication criteria as it currently stands. Therefore, we invite you to submit a revised version of the manuscript that addresses the points raised during the review process.

We look forward to receiving your revised manuscript.

Kind regards,

Ji-Hoon Yun

Academic Editor

PLOS ONE

Journal Requirements:

"This research was funded by the following awards: 1.Industry-University-Research Innovation Foundation of  Chinese University (2021LDA06003,URL:http://www.cutech.edu.cn/cn/index.htm), whose project leader is Huilong Jin. The sponsors play a role in logic structure  and  writing skill of this manuscript.

2. Science and Technology Project of Hebei Education Department (QN2023233, URL: http://jyt.hebei.gov.cn/). whose project leader is Shuang Zhang. The sponsors play a role in 

research direction  and  model design.

3.Science and Technology Research Foundation of Hebei Normal University (L2021B33,URL: https://www.hebtu.edu.cn/) whose project leader is Shuang Zhang. The sponsors play a role in direction of application."

Reviewers' comments:

Reviewer's Responses to Questions

**Comments to the Author**

1. Is the manuscript technically sound, and do the data support the conclusions?

Reviewer #1: Yes

Reviewer #2: Partly

Reviewer #3: No

2. Has the statistical analysis been performed appropriately and rigorously? 

Reviewer #1: N/A

Reviewer #2: N/A

Reviewer #3: No

3. Have the authors made all data underlying the findings in their manuscript fully available?

Reviewer #1: Yes

Reviewer #2: Yes

Reviewer #3: Yes

4. Is the manuscript presented in an intelligible fashion and written in standard English?

Reviewer #1: No

Reviewer #2: No

Reviewer #3: No

5. Review Comments to the Author

Reviewer #1: 1. Replace “un-manned aerial vehicle (UAV)” with “unmanned aerial vehicle (UAV)” in Abstract.

2. Replace “the target optimization value is fast convergence” with “the target optimization value converges fast”.

3. From the statement “is superior to other benchmarks.”. What benchmarks are the authors referring to? Please state them. Furthermore, it is important for the authors to clearly state the improvement values (as a percentage) of the proposed schemes over the baselines.

4. Why choose the exact figure of 300m? Is it about 300m? And the authors should state why such performance in the abstract. In general, the abstract can be better presented.

5. On page 2, replace “in rescue fleetly when disasters occurring” to “in rescue fleet when disaster occurs”.

6. The authors should incorporate closely related works that deploy UAVs in similar scenarios. The authors ignored works that have applied other approaches to address these problems such as interference management and energy efficiency (EE). Therefore, I suggest to the authors consider the following references since they were compared against cluster-based approaches presented in this work:

[A] Communication-Enabled Deep Reinforcement Learning to Optimise Energy-Efficiency in UAV-Assisted Networks, Vehicular Communications, Volume 43.

[B] Multi-Agent Deep Reinforcement Learning For Optimising Energy Efficiency of Fixed-Wing UAV Cellular Access Points," ICC 2022 – IEEE International Conference on Communications, 2022, pp. 1-6,.

7. From the related work section, it’s unclear what the key contribution of the work is. The authors should incorporate a table that highlights each of the closely related work and include those in [A] and [B].

8. The authors will do well to provide the control overhead of the proposed approach with respect to the baselines used in the evaluation.

9. Lots of mathematical notations used. Please provide a list of notation table.

10. We understand that the ground users may be mobile, however, from the system model, the authors consider the deployment of static ground users. How does you algorithm account for the possible change of position of the ground users? Please provide justification for this.

11. How did the authors arrive at the chosen altitude for the fixed-winged UAVs? Authors may provide references to backup their chosen parameters. Ref [B] may be of help.

12. The overall language can be further improved, with proper use of words.

13. The authors have adopted cluster-based approaches, however, it is unclear who performs the clustering task. Moreover, the knowledge of the position of all ground users within the coverage area must be known beforehand. This assumption may not be realistic. The authors should provide a justification for this.

14. Why are some of the results in the reference section. The authors should fix this.

15. Some key parameters are missing from the parameter table. No pathloss exponent, transmit power, clustering metrics used, etc.

16. In scenarios where the users are not evenly distributed in the coverage area, the size of each cluster cannot be fixed and pre-determined. Please provide some justification for this.

Reviewer #2: The manuscript focus on the EE problem for UAV assisted NOMA systems. I have some concerns about the manuscript given below;

1) There are problems with the language of the manuscript. It requires a detailed proofreadimg.

2) Why we require clustering mechanism here?

3) There are problems in the part that explains clustering algorithms.

i) For example in greedy algorithm the readers are directed to the Algorithm 1 and 2 at the same time.

ii) Moreover, the users are not classified as Class A and B in this algorithm, although it is written as " After users were sorted, all users were divided into two kinds of class-A and class-B."

iii) Moreover, in the text, the authors say that "The clustering result is showed in Fig 2, which rectangular blocks of the same color represent the same cluster." but there is no color in the figure.

iv) The Figures 2,3 and 4 require much more explanations. In Figures 3 and 4, why the U3 and U4 are eliminated?

v) In Algorithm 2, why only class B clusters are given? We can not see the class A in the pseudo code.

vi) Algorithm 1, 2 and 3 must be rewritten and must be explained better.

4) In Figure 5b, the trend of the EE is normal? In the EE models, it is expected that EE is saturated after a Pmax value. This part can also be explaned in detail.

5) The EE values seem to low. They are all in bits/Joule. Generally these values are given in Mbits/Joule. Is there any problem for these values?

6) The complexity of the algorithms must also be given. Although, the authors study on the number of iterations, it is not a measure for the complexity of the system. If these complexity issues are given, it will be more fair to compare non-clustering OMA and cluster-based NOMA algorithms. The EE advantage of NOMA is also slightly different compared to OMA. If the complexity of NOMA is higher, this EE gain may be compansated.

7) Please check all equations. For example in Eq. (12), I think it will be R_total (q(t)) instead of R_totalq(t). Also in Eqn. (13) R_totalq(l) is given in the objective function but it is defined as Rtotal [l] in the subjective function. All the notations must be consistent.

Reviewer #3: In this paper, the authors have proposed a NOMA-UAV based energy efficiency scheme and solved the problem based on three clustering algorithms. There are some important issues that should be considered.

1- The writing of the article has many mistakes and problems in content continuity, fluency and explanations and, there are many grammatical errors and wrong statements. Especially in numerical results section. Thus all parts should be rewritten.

2- The novelty of the paper is low. Only, in the problem solution, authors use the combination of two spectrum allocations scheme named hybrid clustering. .

3- In SIC system, to interference cancelation, assigned power to the user should be allocated according to the channels gain, thus, this constraint should be applied in the problem, please refer to the reference 31.

4- In equation 12, C3 can cover C2.

5- In line 180, authors assume no-intra interference among user. Thus, no need to use NOMA.

6- In section 4.1.1, what is the meaning of the paragraph 2?

7- The explanations of the Fig 6 are not clear and reviewer thinks Fig.6 has no information.

8- In Fig 5a, sum rate is around 815 bits for all NOMA scheme and there are no superior differences between them.

6. PLOS authors have the option to publish the peer review history of their article (what does this mean?). If published, this will include your full peer review and any attached files.

Reviewer #1: No

Reviewer #2: No

Reviewer #3: No

---

## [Author Response · Author response to Decision Letter 0]

7 Oct 2023

Dear Editor and Reviewers:

On behalf of my co-authors, we are very grateful to you for giving us the opportunity to revise our manuscript. We appreciate you very much for your positive and constructive comments and suggestions on our manuscript entitled “A NOMA Scheme with Resource Allocation and Trajectory Optimization” (PONE-D-23-21957).

We have studied the comments of all Reviewers carefully and tried our best to revise our manuscript according to the comments. The file "Response to Reviewers" is the responses and revisions. We have made in response to the questions and suggestions on an item-by-item basis. Thanks again to the hard work of the editor and reviewers.

---

## [Decision Letter · Decision Letter 1]

2 Nov 2023

PONE-D-23-21957R1Energy-Efficient UAV Communication: A NOMA Scheme with Resource Allocation and Trajectory OptimizationPLOS ONE

Dear Dr. Zhang,

Thank you for submitting your manuscript to PLOS ONE. After careful consideration, we feel that it has merit but does not fully meet PLOS ONE’s publication criteria as it currently stands. Therefore, we invite you to submit a revised version of the manuscript that addresses the points raised during the review process.

We look forward to receiving your revised manuscript.

Kind regards,

Ji-Hoon Yun

Academic Editor

PLOS ONE

Reviewers' comments:

Reviewer's Responses to Questions

**Comments to the Author**

1. If the authors have adequately addressed your comments raised in a previous round of review and you feel that this manuscript is now acceptable for publication, you may indicate that here to bypass the “Comments to the Author” section, enter your conflict of interest statement in the “Confidential to Editor” section, and submit your "Accept" recommendation.

Reviewer #1: (No Response)

Reviewer #2: (No Response)

Reviewer #3: (No Response)

2. Is the manuscript technically sound, and do the data support the conclusions?

Reviewer #1: Partly

Reviewer #2: (No Response)

Reviewer #3: Partly

3. Has the statistical analysis been performed appropriately and rigorously? 

Reviewer #1: No

Reviewer #2: (No Response)

Reviewer #3: Yes

4. Have the authors made all data underlying the findings in their manuscript fully available?

Reviewer #1: Yes

Reviewer #2: (No Response)

Reviewer #3: Yes

5. Is the manuscript presented in an intelligible fashion and written in standard English?

Reviewer #1: (No Response)

Reviewer #2: (No Response)

Reviewer #3: No

6. Review Comments to the Author

Reviewer #1: 1. The authors have not been able to address several comments highlighted in the first review. The authors should address Comment 8, where they do not provide the control overhead for the proposed algorithm.

2. From Comment 11, the authors have set the height increment for each flight to 50m. How is this possible and under what consideration? This is highly impractical as it may jeopardise the findings in the results. The authors may have to reconsider their choice of 50m as this may affect the validity of the results.

3. The authors are answering a different question from what was being asked in Comment 14 and 15.

Reviewer #2: It is difficult to follow if the authors addressed all questions of the reviewers. The revised parts in the manuscripts must be emphasized in the answers document. For example, the explanation has been added on Page XX line xxx.

Reviewer #3: 1- Although the authors have improved the article and added many explanations unfortunately the article still is not fluent and there are lots of grammatical mistake especially in new added parts.

2- As the novelty of the work is low, according to table 1. It is necessary that the performance of the work is compared with those in the some references such as 24 and 26 .

3- What is 'm' in the C4 equation?

7. PLOS authors have the option to publish the peer review history of their article (what does this mean?). If published, this will include your full peer review and any attached files.

Reviewer #1: No

Reviewer #2: No

Reviewer #3: No

---

## [Author Response · Author response to Decision Letter 1]

2 Dec 2023

On behalf of my co-authors, we are very grateful to you for giving us the last opportunity to revise our manuscript. We appreciate you very much for your positive and constructive comments and suggestions on our manuscript entitled “A NOMA Scheme with Resource Allocation and Trajectory Optimization” (PONE-D-23-21957).

We have studied the comments of all Reviewers carefully and tried our best to revise our manuscript according to the comments. The file "Response to Reviewers 2" are the responses and revisions. We have made in response to the questions and suggestions on an item-by-item basis. Thanks again to the hard work of the editor and reviewers.

---

## [Decision Letter · Decision Letter 2]

26 Dec 2023

PONE-D-23-21957R2Energy-Efficient UAV Communication: A NOMA Scheme with Resource Allocation and Trajectory OptimizationPLOS ONE

Dear Dr. Zhang,

Thank you for submitting your manuscript to PLOS ONE. After careful consideration, we feel that it has merit but does not fully meet PLOS ONE’s publication criteria as it currently stands. Therefore, we invite you to submit a revised version of the manuscript that addresses the points raised during the review process.

We look forward to receiving your revised manuscript.

Kind regards,

Ji-Hoon Yun

Academic Editor

PLOS ONE

Reviewers' comments:

Reviewer's Responses to Questions

**Comments to the Author**

1. If the authors have adequately addressed your comments raised in a previous round of review and you feel that this manuscript is now acceptable for publication, you may indicate that here to bypass the “Comments to the Author” section, enter your conflict of interest statement in the “Confidential to Editor” section, and submit your "Accept" recommendation.

Reviewer #1: All comments have been addressed

Reviewer #2: (No Response)

Reviewer #3: (No Response)

2. Is the manuscript technically sound, and do the data support the conclusions?

Reviewer #1: Yes

Reviewer #2: (No Response)

Reviewer #3: Yes

3. Has the statistical analysis been performed appropriately and rigorously? 

Reviewer #1: Yes

Reviewer #2: (No Response)

Reviewer #3: Yes

4. Have the authors made all data underlying the findings in their manuscript fully available?

Reviewer #1: Yes

Reviewer #2: (No Response)

Reviewer #3: Yes

5. Is the manuscript presented in an intelligible fashion and written in standard English?

Reviewer #1: Yes

Reviewer #2: (No Response)

Reviewer #3: No

6. Review Comments to the Author

Reviewer #1: The authors have addressed all my comments. I believe the paper is in a much better form. Hence, I recommend that it be accepted. Thanks.

Reviewer #2: The authors did not answer all questions point by point. They seem to prefer select some of them to answer, that is not a good way.

Reviewer #3: Authors have revised the article and almost answered the questions but there are some problems in the structure and statements of the article. For example:

1- Last statement of the abstract.

2- Statements in line 9-10 , in line 34-35 and in line 42-43 in introduction section

Furthermore, there are many mistakes in other parts of the article that should be revised.

7. PLOS authors have the option to publish the peer review history of their article (what does this mean?). If published, this will include your full peer review and any attached files.

Reviewer #1: No

Reviewer #2: No

Reviewer #3: No

---

## [Author Response · Author response to Decision Letter 2]

31 Jan 2024

Dear Reviewers:

On behalf of my coauthors, I am grateful to you for giving us this opportunity to revise our manuscript. We appreciate your positive and constructive comments and suggestions on our manuscript titled “A NOMA Scheme with Resource Allocation and Trajectory Optimization” (PONE-D-23-21957).

We have studied the comments of all the Reviewers carefully and tried our best to revise our manuscript accordingly. The following summarizes the responses and revisions. We have responded to the questions and suggestions on an item-by-item basis in the file "Response to Reviewers 3" . Thank you again for the hard work of the editor and reviewers.

---

## [Editor Report · Decision Letter 3]

5 Mar 2024

PONE-D-23-21957R3Energy-Efficient UAV Communication: A NOMA Scheme with Resource Allocation and Trajectory OptimizationPLOS ONE

Dear Dr. Zhang,

Thank you for submitting your manuscript to PLOS ONE. After careful consideration, we feel that it has merit but does not fully meet PLOS ONE’s publication criteria as it currently stands. Therefore, we invite you to submit a revised version of the manuscript that addresses the points raised during the review process. Based on the reviewers' comments and my own reading of your manuscript, I recommend that you update your paper and submit a revised version for further processing. Please include a detailed letter outlining point-by-point changes corresponding to each of the reviewers' comments. It is particularly important to address the comments raised by Reviewer 2 in the first-round review. Be sure to provide clear references to the sections, pages, and line numbers where changes have been made in your responses.

We look forward to receiving your revised manuscript.

Kind regards,

Ji-Hoon Yun

Academic Editor

PLOS ONE

---

## [Author Response · Author response to Decision Letter 3]

12 Mar 2024

Thank you very much for your attention to the manuscript. This time, we will only provide a detailed response to the comments from Reviewer 2, and the revised manuscript with track changes accordingly for Reviewer 2. The relevant response has been described in the file "Response to Reviewers&#39". Thank you for your patient support and comment with our work again.

---

## [Editor Report · Decision Letter 4]

24 Mar 2024

Energy-Efficient UAV Communication: A NOMA Scheme with Resource Allocation and Trajectory Optimization

PONE-D-23-21957R4

Dear Dr. Zhang,

We’re pleased to inform you that your manuscript has been judged scientifically suitable for publication and will be formally accepted for publication once it meets all outstanding technical requirements.

Kind regards,

Ji-Hoon Yun

Academic Editor

PLOS ONE

Additional Editor Comments (optional):

All the concerns raised by Reviewer 2 have been appropriately addressed in the revised manuscript, and detailed responses to each point have been effectively summarized in the revision letter.

---

## [Editor Report · Acceptance letter]

3 Apr 2024

PONE-D-23-21957R4 

PLOS ONE

Dear Dr. Zhang, 

I'm pleased to inform you that your manuscript has been deemed suitable for publication in PLOS ONE. Congratulations! Your manuscript is now being handed over to our production team.

Kind regards, 

on behalf of

Dr. Ji-Hoon Yun 

Academic Editor

PLOS ONE